# Reactive Control for Collision Evasion with Extended Obstacles

**DOI:** 10.3390/s22155478

**Published:** 2022-07-22

**Authors:** Jonghoek Kim

**Affiliations:** Electronic and Electrical Department, Sungkyunkwan University, Suwon 03063, Korea; jonghoek@gmail.com

**Keywords:** reactive collision evasion, underwater environment, maximum acceleration, extended obstacle, active sonar sensor

## Abstract

Evading collisions in three-dimensional underwater environments is critical in exploration of an Autonomous Underwater Vehicle (AUV). In underwater environments, AUV measures an obstacle surface by utilizing a three-dimensional active sonar. This article addresses reactive collision evasion control by considering extended obstacles. Here, an extended obstacle is an arbitrary obstacle that can generate any number of measurements and not a point target generating at most one measurement. Considering 3D environments, our manuscript considers collision evasion with both moving obstacles and static obstacles. The proposed reactive collision evasion controllers are developed by considering hardware limits, such as the maximum speed or acceleration limit of an AUV. We further address how to make an AUV move towards a goal, while avoiding collision with extended obstacles. As far as we know, the proposed collision evasion controllers are novel in handling collision avoidance with an extended obstacle, in the case where an AUV measures 3D-obstacle boundaries by utilizing sonar sensors. The effectiveness of the proposed controllers is demonstrated by MATLAB simulations.

## 1. Introduction

Recently, autonomous underwater vehicles (AUVs) have been utilized in many applications, such as underwater exploration. Collision evasion is crucial for safe maneuvers of AUVs. In underwater environments, electromagnetic signals dissipate quickly. Thus, an AUV uses sonar sensors with a limited range in order to detect nearby underwater objects. This article addresses reactive controls that do not require path planning ahead of time. The reactive collision evasion control is developed by considering limited sensors and the hardware limits of AUV.

We consider the case where an AUV uses active sonar sensors to sense its surroundings. We assume that the AUV has sonar sensors for calculating the point cloud of underwater objects [1]. The authors of [1] proposed a method to reconstruct an object’s three-dimensional geometry based on sonar images captured while an AUV moves over the object. By reconstructing the 3D shape of the object, [1] further addressed a method for classifying the object using neural networks. In our paper, point clouds of an object are utilized for collision evasion control.

In our manuscript, the AUV can calculate three-dimensional points, termed *objectPoints*, on object surfaces. The proposed reactive controllers make AUV evade collision with every objectPoint.

The AUV in our article uses sonar sensors to calculate the point cloud of underwater objects [1]. AUV measures an obstacle surface using sonar sensors, followed by estimating the obstacle’s position and velocity using the information of objectPoints. By measuring the difference between two clouds of objectPoints measured at distinct sampling indexes, the extended obstacle’s velocity can be estimated. In order to estimate the velocity (speed and orientation) of objectPoints on an extended obstacle, AUV utilizes iterative closest point (ICP) algorithms [2,3,4,5,6,7]. In order to limit the influence of outliers, we use the robust ICP algorithm in [6]. Moreover, for efficient search, this paper uses k-D tree for matching processes [8].

There are many papers on developing collision evasion controls in two-dimensional environments [9,10,11,12]. In [13], the obstacle avoidance research of wave glider in 2D marine environment was conducted. Considering 2D environments, artificial potential field (APF) has been widely used for avoid collision between a robot and obstacles [13,14]. APF is generated by the addition of the attraction potential field (generated by the goal) and the repulsive potential field (generated by obstacles).

As far as we know, APF is not provably complete in achieving collision avoidance. For instance, consider the case where the robot needs to move into a tunnel for reaching its goal. Due to the repulsive field generated by obstacles at the tunnel entrance, the robot has difficulty in maneuvering through the tunnel. Moreover, APF was not applied for collision avoidance in 3D environments. Moreover, APF is not suitable for avoiding moving obstacles, since the velocity of an obstacle was not considered in APF. Considering 3D environments, our article tackles collision evasion with both maneuvering obstacles and static obstacles.

Velocity obstacle (VO) methods were utilized for collision evasion with obstacles, which may move [9,11,15,16,17]. VO methods are computationally efficient and, thus, appropriate for mobile robots [18]. VO methods are provably complete in achieving collision avoidance, as long as an obstacle maintains its velocity for all future times [15]. VO methods assume that an obstacle moves with a constant velocity for all future times. However, this is not a valid assumption in practice.

In our manuscript, the proposed reactive collision avoidance controls are developed and inspired by VO methods. We address 3D reactive collision avoidance controls, assuming that an obstacle moves with a constant velocity within $k_{p}$ sampling-indexes in the future. At each sampling-index, an obstacle’s pose (position and velocity) is estimated using sonar sensor measurements, and we run collision avoidance controls at every sampling-index. Our study proves that as long as the obstacle velocity (speed and orientation) is estimated correctly using ICP algorithms, collision avoidance is assured within $k_{p}$ sampling indexes in the future.

Considering 3D environments, our manuscript handles collision evasion with both moving obstacles and static obstacles. Several papers [19,20,21] considered collision evasion in three-dimensional environments. The authors of [20] presented a collision avoidance algorithm based on potential fields for fixed-wing unmanned aerial vehicles (UAVs) with constrained field-of-view (FOV) sensors such as cameras. The authors of [22] explored 3D path planning for UAVs in 3D point-cloud environments. The approach in [22] searched for obstacle-free and smooth paths by analyzing a point cloud of the target environment, using a modified rapidly exploring random tree (RRT)-based path planning algorithm. However, collision avoidance controllers in [20,21,22] are not suitable for evading a fast moving obstacle, since they did not consider the velocity of a maneuvering obstacle.

References [19,23,24,25,26] developed collision avoidance controllers while modeling obstacles as spheres or bounding boxes. The authors of [23] addressed a reactive algorithm for avoiding obstacles in three-dimensional space by modeling obstacles as spheres.

In our article, we do not model obstacles as spheres or bounding boxes. Our study handles collision evasion with extended obstacles (moving obstacles as well as static obstacles). Here, an extended obstacle is an arbitrary obstacle generating any number of measurements and not a point target generating at most one measurement. To the best of our knowledge, the proposed collision evasion controllers are novel, since our manuscript handles the case where an AUV measures three-dimensional obstacle surfaces by utilizing sonar sensors. In our paper, the AUV measures an obstacle surface and estimates its position and velocity using the information of objectPoints on the surface.

Considering 3D environments, our collision avoidance controls avoid moving obstacles as well as static obstacles. The proposed reactive controllers utilize the velocity information of an obstacle; hence, it is suitable for evading a fast-moving obstacle. In practice, AUVs may need to approach its goal while avoiding obstacles. We thus address a method to make AUVs approach its goal, while avoiding collision with extended obstacles.

The proposed reactive collision-evasion controllers are developed by considering an AUV’s hardware limits, such as the maximum speed or the acceleration limit of the AUV. At each sampling-index *k*, the AUV calculates its velocity command, which allows it to approach the goal, while avoiding collision with extended obstacles (maneuvering obstacles as well as static obstacles). The velocity command is set considering hardware limits, such as the maximum speed or acceleration limit of the AUV.

Our article handles the case where AUV measures three-dimensional obstacle surfaces using sonar sensors. To the best of our knowledge, the proposed collision evasion controllers are novel, since our article tackles collision avoidance with an extended obstacle. The outperformance of the proposed evasion controllers is demonstrated by utilizing MATLAB simulations.

Section 2 discusses the preliminaries of this article. Section 3 handles assumptions and definitions of this article. Section 4 handles how to estimate the velocity (speed and orientation) of objectPoints associated with an extended obstacle. Section 5 addresses the proposed reactive collision-evasion control. Section 6 discusses how to make AUVs move towards the goal, while avoiding collision with extended obstacles. Section 7 introduces MATLAB simulations to demonstrate the performance of the proposed controls. Section 8 addresses our conclusions.

## 2. Preliminaries

This article utilizes two frames: an inertial frame {I} and a body frame {B} [27]. We address several definitions in rigid-body dynamics [27].

The origin of {I} is a point with three axes pointing north, east, and south. {B} is fixed to AUV’s body such that {B} is originated at AUV’s gravity center.

In rigid-body dynamics [27], θ and ψ define *pitch* and *yaw*, respectively. For notation convenience, let c(η) represent cos(η). In addition, let s(η) represent sin(η). Let t(η) represent tan(η).

The rotation matrix representing the counterclockwise (CC) rotation of an angle ψ centered at the *z*-axis in {B} is as follows.
(1)$$\begin{array}{c} r\left(\psi \right)=\left(\begin{array}{ccc} c(\psi ) & -s(\psi ) & 0\\ s(\psi ) & c(\psi ) & 0\\ 0 & 0 & 1 \end{array}\right). \end{array}$$

The rotation matrix representing the CC rotation of an angle θ centered at the *y*-axis in {B} is as follows.
(2)$$\begin{array}{c} r\left(\theta \right)=\left(\begin{array}{ccc} c(\theta ) & 0 & s(\theta )\\ 0 & 1 & 0\\ -s(\theta ) & 0 & c(\theta ) \end{array}\right). \end{array}$$

The combined rotation matrix is built by multiplying (Equation 1) and (Equation 2) to obtain the following.
(3)$$\begin{array}{c} r(\psi ,\theta )=r(\psi )r(\theta ). \end{array}$$

## 3. Definitions and Assumptions

This article assumes that the localization of AUV is performed using AUV’s on-board sensors, such as the inertial measurement unit (IMU). The question of how to localize AUV is not within the scope of this article. Various methods [28,29,30] exist for the localization of an AUV.

We introduce teh notations used in this paper. Let $A\left(v_{1},v_{2}\right)=acos\left(\frac{v_{1}\cdot v_{2}}{∥v_{1}∥∥v_{2}∥}\right)$ define the angle between two three-dimensional vectors $v_{1}$ and $v_{2}$. Moreover, L(A and B) indicates the line segment for which its two end points are A and B, respectively.

Let $q\in R^{3}$ present the three-dimensional coordinates of the AUV. Let $v=\dot{q}\in R^{3}$ present the AUV’s velocity. Let $h=\frac{v}{∥v∥}\in R^{3}$ define the *orientation vector* of the AUV.

We consider discrete-time systems. Considering a variable A, A(k) indicates A at sampling-index *k*. For instance, AUV at sampling-index *k* is located at $q\left(k\right)\in R^{3}$.

Suppose that the current sampling-index is *k*. The process model of AUV is as follows:(4)$$\begin{array}{c} q(k+1)=q(k)+v(k)*dt, \end{array}$$
where dt indicates the sample duration in discrete-time systems. In (Equation 4), v(k) can be considered as the velocity command (control input) of the AUV at sampling-index *k*. The motion model in (Equation 4) is commonly used in reactive collision avoidance using VO methods [9,11,15,16,17].

Inspired by VO methods [9,11,15,16,17], we search for the velocity command, v(k), which satisfies collision avoidance at each sampling-index *k*. At each sampling-index *k*, the velocity command v(k) is set by considering hardware limits, such as the maximum speed or the acceleration limit of the AUV. The AUV’s maximum speed is $s_{max}$, i.e., $∥v\left(k\right)∥\leq s_{max}$. The maximum acceleration rate of AUV is $a_{max}$. In other words, we have the following.
(5)$$\begin{array}{c} -a_{max}\leq \frac{\left(∥v(k+1)∥-∥v(k)∥\right)}{dt}\leq a_{max}. \end{array}$$

In practice, the AUV turns with a bounded turn rate. Considering the bounded turn rate of the AUV, this article uses the following:(6)$$\begin{array}{c} A(v(k),v(k+1))<\alpha *dt \end{array}$$
at each sampling-index *k*.

At each sampling-index *k*, we calculate the velocity command v(k) in (Equation 4), which allows the AUV to approach a goal, while avoiding collisions with extended obstacles (moving obstacles as well as static obstacles). Velocity command v(k) in (Equation 4) can be considered as a high-level control command of the AUV. Once v(k) is set at sampling-index *k*, the desired waypoint at sampling-index k+1 is set as q(k+1) in (Equation 4). Then, the straight line segment from the current AUV’s position q(k) to the next waypoint q(k+1) is set as the reference trajectory from sampling-index *k* to k+1. Trajectory tracking controls in [31,32,33,34] can be applied to make the AUV track the straight-line segment trajectory, even in environments with model dynamic uncertainties and the presence of external disturbances representing ocean currents and waves. Note that detailed kinematic model in AUV’s local coordinate frame is handled in the trajectory tracking controls in [31,32,33,34].

At each sampling-index *k*, we calculate velocity command v(k) in (Equation 4), assuming that an obstacle moves with a constant velocity within $k_{p}$ sampling indexes in the future. At each sampling index, an obstacle’s pose (position and velocity) is estimated. In addition, we run collision avoidance controls at every sampling-index.

The AUV has sonar sensors to derive the point cloud of underwater objects [1]. The AUV measures three-dimensional points on obstacle surfaces utilizing three-dimensional sonar sensors. Recall that every three-dimensional point is termed objectPoint, say $o_{p}$. Rendering the AUV’s conjecture as an extended obstacle’s velocity (speed and orientation) using objectPoints is addressed in Section 4.

We say that the AUV and an objectPoint $o_{p}$ is in a *near-collision state* as the relative distance between them is decreases than compared to a certain constant, say r>0. Here, r>0 is set by considering the size of the AUV. In other words, a sphere with radius *r* is sufficiently large to contain the entire AUV. Let $S_{r}\left(o_{p}\right)$ define the sphere with radius *r*, for which its center is at $o_{p}$.

## 4. Conjecture the Velocity of ObjectPoints on an Extended Obstacle

To conjecture the velocity of objectPoints on an extended obstacle at sampling-index *k*, the AUV utilizes a robust ICP algorithm [6]. Let *k* present the current sampling index. Let S(k) define the set of objectPoints measured at sampling-index *k*.

Suppose that S(k) is given by $S\left(k\right)=\left\{x_{i}\right\}$, $i\in \{1,\ldots ,M_{k}\}$ Moreover, suppose that $S(k-k_{d})$ is given by $S\left(k-k_{d}\right)=\left\{y_{j}\right\}$, $j\in \{1,\ldots ,M_{k-k_{d}}\}$. Here, $k_{d}>0$ is a tuning constant. This indicates that the AUV measures $M_{k}$ objectPoints at sampling-index *k*. In addition, the AUV measures $M_{k-k_{d}}$ objectPoints at sampling-index $k-k_{d}$. Observe that $x_{i},y_{j}\in R^{3}$ are three-dimensional objectPoint coordinates.

Suppose that both $M_{k}$ and $M_{k-k_{d}}$ exceed a certain threshold, say $Th_{ICP}$. Then, the ICP algorithm is applied to both S(k) and $S(k-k_{d})$ in order to find a velocity vector of the obstacle at sampling-index *k*.

The ICP algorithm at sampling-index *k* estimates a rigid motion with rotation matrix $R_{k}$ and translation $t_{k}$, which minimizes the following error *E*:(7)$$\begin{array}{c} E\left(R_{k},t_{k}\right)=\sum_{i=1}^{M_{k}}w_{i}{∥\left(R_{k}x_{i}+t_{k}-y_{j^{*}}\right)∥}^{2}, \end{array}$$
where $w_{i}$ is the weight for the *i*-th cloud point and is selected considering Huber’s robust function [6,7]. The method for selecting $w_{i}$ is addressed in [6,7].

Given $R_{k}$ and $t_{k}$, point $y_{j^{*}}\in S\left(k-k_{d}\right)$ is denoted as the optimal correspondence of $x_{i}$, which is the closest point to the transformed $x_{i}\in S\left(k\right)$.
(8)$$\begin{array}{c} j^{*}=argmin_{j\in \{1,2,\ldots ,M_{k-k_{d}}\}}{∥\left(R_{k}x_{i}+t_{k}-y_{j}\right)∥}^{2}. \end{array}$$

Given an initial transformation $\left(R_{k},t_{k}\right)$, the ICP algorithm iteratively solves the problem by alternating between estimating the transformation with (Equation 7), and finding the closest-point matches with (Equation 8). For an efficient search, the k-D tree is utilized for the matching process [8]. This iterative process guarantees convergence to a local minimum. Once the ICP algorithm is completed at sampling-index *k*, translation $t_{k}$ indicates the velocity of the maneuvering obstacle at sampling-index *k*.

Note that the ICP algorithm is an iterative algorithm and, thus, requires an initial transformation $\left(R_{k},t_{k}\right)$. We next present a method for setting an initial transformation $\left(R_{k},t_{k}\right)$. The rotation matrix $R_{k}$ is initialized as a diagonal matrix. Moreover, the translation vector $t_{k}$ is initialized as follows:(9)$$\begin{array}{c} t_{k}=\frac{\sum_{j=1}^{M_{k-k_{d}}}y_{j}}{M_{k-k_{d}}}-\frac{\sum_{i=1}^{M_{k}}x_{i}}{M_{k}}. \end{array}$$

There may be a case where $\left(R_{k},t_{k}\right)$ is not estimated correctly due to the mismatch of point pairs. We assume that the maximum speed of an obstacle is known a priori. If $∥t_{k}∥$ exceeds the maximum speed of an obstacle, then we set the following:(10)$$\begin{array}{c} t_{k}=t_{k-1}, \end{array}$$
which implies that we use the translation, which was estimated at the previous sampling index.

## 5. Reactive Controllers for Collision Evasion

### 5.1. Collision Prediction

At each sampling-index *k*, we calculate the velocity command v(k) in (Equation 4). Let v denote the velocity command v(k) for convenience.

At each sampling-index *k*, the AUV calculates v, assuming that the objectPoint $o_{p}$ maneuvers with a velocity $v_{O}$ for $k_{p}$ sampling indexes in the future. Recall that Section 4 discussed how to conjecture $v_{O}$ based on the sonar measurements of AUV. At each sampling-index, the position and velocity of $o_{p}$ are estimated. Furthermore, one runs collision avoidance controls at every sampling index.

We discuss how to evade the case where AUV and an objectPoint, say $o_{p}$, collide between sampling-index *k* and $k+k_{p}$. No collision happens between sampling-index *k* and $k_{p}$ if
(11)$$\begin{array}{c} ∥q\left(k\right)-o_{p}\left(k\right)+\left(v-v_{O}\right)u*dt∥>r. \end{array}$$
for all $0\leq u\leq k_{p}$. As one increases *u* in (Equation 11) from 0 to $k_{p}$, $q\left(k\right)+\left(v-v_{O}\right)u*dt$ generates discrete points along line segment $L(q\left(k\right),q\left(k\right)+\left(v-v_{O}\right)*k_{p}*dt)$.

Utilizing (Equation 11), Lemma 1 is derived. This lemma proves that as long as the obstacle velocity $v_{O}$ is estimated correctly using ICP algorithms in Section 4, collision avoidance between $o_{p}$ and AUV is assured within $k_{p}$ sampling indexes in the future.

**Lemma** **1.**
*An objectPoint $o_{p}$ maneuvers with a velocity $v_{O}$ for $k_{p}$ sampling indexes in the future. In addition, the AUV maneuvers with a velocity v. Assume that q(k) is outside $S_{r}\left(o_{p}\right)$ at sampling-index k. The AUV does not collide with $o_{p}$ for $k_{p}$ sampling-indexes in the future, as long as $L(q\left(k\right),q\left(k\right)+\left(v-v_{O}\right)*k_{p}*dt)$ does not meet $S_{r}\left(o_{p}\right)$ at sampling-index k.*


The velocities of an object and that of AUV are considered in collision avoidance, as presented in Lemma 1. Using Lemma 1, the AUV can predict its collision within $k_{p}$ sampling-indexes in the future. At each sampling-index *k*, th eAUV examines whether its velocity command v meets the following *non-collision requirement*: *For every objectPoint $o_{p}$, $L(q\left(k\right),q\left(k\right)+\left(v-v_{O}\right)*k_{p}*dt)$ does not meet $S_{r}\left(o_{p}\right)$ at sampling-index k.*

If $L(q\left(k\right),q\left(k\right)+\left(v-v_{O}\right)*k_{p}*dt)$ does not meet $S_{r}\left(o_{p}\right)$, then the AUV utilizes v as its velocity command for a single sample duration dt. If $L(q\left(k\right),q\left(k\right)+\left(v-v_{O}\right)*k_{p}*dt)$ meets $S_{r}\left(o_{p}\right)$ for any objectPoint $o_{p}$, then the AUV needs to find a non-collision velocity vector, as addressed in Section 5.2.

### 5.2. Find a Non-Collision Velocity Vector at Sampling-Index k+1

As the non-collision requirement is not fulfilled for at least one objectPoint, the AUV needs to find a velocity command, v, satisfying the non-collision requirement. Once a velocity command, which satisfies the non-collision requirement, is found, then the AUV utilizes the found velocity as its velocity command for a single sample duration dt. Note that, at every sampling-index, we run collision avoidance controls.

Algorithm 1 shows how to find a non-collision velocity at sampling-index k+1 utilizing (Equation 17). Let $u={\left(1,0,0\right)}^{T}$ present the orientation of the vehicle in the body frame. Under (Equation 3), one derives the following.
(12)$$\begin{array}{c} v(k)=r(\psi (k),\theta (k))*∥v(k)∥*u. \end{array}$$

We control only the pitch and the yaw of the body in order to achieve the desired orientation. We, thus, find the body orientation, say ψ(k),θ(k), associated with v(k). Under (Equation 12), one calculates the following:(13)$$\begin{array}{c} h\left(k\right)={\left(h\left(k,1\right),h\left(k,2\right),h\left(k,3\right)\right)}^{T}, \end{array}$$
where $h\left(k\right)=\frac{v(k)}{∥v(k)∥}$ indicates AUV’s orientation vector at sampling-index *k*. Furthermore, h(k,1)=c(ψ(k))∗c(θ(k)), h(k,2)=s(ψ(k))∗c(θ(k)), and h(k,3)=−s(θ(k)). Here, h(k,j) indicates the *j*-th element of h(k).

Under (Equation 13), one calculates the following.
(14)$$\begin{array}{c} \theta \left(k\right)=atan2(-h\left(k,3\right),\sqrt{h{\left(k,1\right)}^{2}+h{\left(k,2\right)}^{2}}). \end{array}$$

Under (Equation 13), one calculates ψ(k). If c(θ(k))>0, then one utilizes the following.
(15)$$\begin{array}{c} \psi (k)=atan2(h(k,2),h(k,1)). \end{array}$$

If c(θ(k))<0, then one utilizes the following.
(16)$$\begin{array}{c} \psi (k)=atan2(-h(k,2),-h(k,1)). \end{array}$$

Recall that the AUV’s speed is limited by $s_{max}$ and that $∥\dot{v}∥\leq a_{max}$. Considering these limits, ∥v(k+1)∥ is limited as follows.
(17)$$\begin{array}{c} A\leq ∥v(k+1)∥\leq A+B. \end{array}$$

Here, $A=max(∥v\left(k\right)∥-a_{max}*dt,s_{min})$ and $B=min(∥v\left(k\right)∥+a_{max}*dt,s_{max})-(∥v\left(k\right)∥-a_{max}*dt)$. In addition, $s_{min}$ is the minimum speed of the AUV, which is used to avoid the case where the AUV stops moving.

Algorithm 1 finds a non-collision speed utilizing the following FOR loop: for $v_{0}=A$, $v_{0}=v_{0}+\frac{B}{N}$, while $v_{0}\leq A+B$. This FOR statement indicates that one examines a velocity vector with low speeds before examining a velocity vector with high speeds. In this manner, the AUV does not speed up abruptly during its maneuver. Note that a low speed is desirable, considering the safety of both the AUV and a moving object, such as other AUVs.
**Algorithm 1** Find a non-collision velocity command at sampling-index k+1 (change yaw only)1:The current sampling-index is *k*;2:**for**$v_{0}=A$, $v_{0}=v_{0}+\frac{B}{N}$, while $v_{0}\leq A+B$**do**3:   $V_{c}^{I}=R\left(\psi \left(k\right),\theta \left(k\right)\right)*v_{0}*u$;4:   **if** col($V_{c}^{I}$, objectSet)==0; **then**5:     Return $V_{c}^{I}$ as the velocity command, and this algorithm is finished;6:   **end if**7:   **for** $\psi_{0}=-\alpha *dt$, $\psi_{0}=\psi_{0}+\frac{\alpha *dt}{N}$, while $\psi_{0}<\alpha *dt$ **do**8:     $V_{c}^{I}=R\left(\psi \left(k\right),\theta \left(k\right)\right)*R\left(\psi_{0}\right)*v_{0}*u$;9:     **if** col($V_{c}^{I}$, objectSet)==0; **then**10:        Return $V_{c}^{I}$ as the velocity command, and this algorithm is finished;11:     **end if**12:     $V_{c}^{I}=R\left(\psi \left(k\right),\theta \left(k\right)\right)*R\left(-\psi_{0}\right)*v_{0}*u$;13:     **if** col($V_{c}^{I}$, objectSet)==0; **then**14:        Return $V_{c}^{I}$ as the velocity command, and this algorithm is finished;15:     **end if**16:   **end for**17:**end for**18:Run Algorithm 2 (change both yaw and pitch);

Algorithm 1 finds a non-collision velocity command while the AUV does not change its pitch. In this manner, the AUV’s unnecessary depth change can be avoided. The AUV only finds a non-collision velocity while changing its yaw angle only.

Algorithm 1 finds a non-collision speed utilizing the following FOR loop: for $\psi_{0}=-\alpha *dt$, $\psi_{0}=\psi_{0}+\frac{\alpha *dt}{N}$, while $\psi_{0}<\alpha *dt$. The right turn is examined before the left turn in Algorithm 1. This order is set by considering the collision regulations (COLREGS) [35]. Suppose that there is a maneuvering object, such as other AUV, which also obeys COLREGs. As both the moving object and AUV obey COLREGs, they can avoid collisions with each other when they are close to each other.

In Algorithm 1, objectSet indicates the set of objectPoint sensed at sampling-index *k*. If $V_{c}^{I}$ and every objectPoint in objectSet meet the non-collision requirement, then this implies that $V_{c}^{I}$ is a non-collision velocity. Thus, we set $V_{c}^{I}$ as the velocity vector at sampling-index k+1, and Algorithm 1 is completed.
**Algorithm 2** Find a non-collision velocity command at sampling-index k+1 (change both yaw and pitch)1:The current sampling-index is *k*;2:minColNum=∞;3:**for**$v_{0}=A$, $v_{0}=v_{0}+\frac{B}{N}$, while $v_{0}\leq A+B$**do**4:   **for**
$\theta_{0}=-\alpha *dt$, $\theta_{0}=\theta_{0}+\frac{2*\alpha *dt}{N}$, while $\theta_{0}<\alpha *dt$ **do**5:     $V_{c}^{I}=R\left(\psi \left(k\right),\theta \left(k\right)\right)*R\left(\theta_{0}\right)*v_{0}*u$;6:     $colNum=col(V_{c}^{I},objectSet)$;7:     **if** colNum==0; **then**8:        Return $V_{c}^{I}$ as the velocity command, and this algorithm is finished;9:     **else**10:        updateMin($colNum,minColNum,0,\theta_{0},v_{0}$) in Algorithm 3;11:     **end if**12:     **for** $\psi_{0}=-\alpha *dt$, $\psi_{0}=\psi_{0}+\frac{\alpha *dt}{N}$, while $\psi_{0}<\alpha *dt$ **do**13:        $V_{c}^{I}=R\left(\psi \left(k\right),\theta \left(k\right)\right)*R\left(\theta_{0}\right)*R\left(\psi_{0}\right)*v_{0}*u$;14:        $colNum=col(V_{c}^{I},objectSet)$;15:        **if** colNum==0; **then**16:          Return $V_{c}^{I}$ as the velocity command, and this algorithm is finished;17:        **else**18:          updateMin($colNum,minColNum,\psi_{0},\theta_{0},v_{0}$) in Algorithm 3;19:        **end if**20:        $V_{c}^{I}=R\left(\psi \left(k\right),\theta \left(k\right)\right)*R\left(\theta_{0}\right)*R\left(-\psi_{0}\right)*v_{0}*u$;21:        $colNum=col(V_{c}^{I},objectSet)$;22:        **if** colNum==0; **then**23:          Return $V_{c}^{I}$ as the velocity command, and this algorithm is finished;24:        **else**25:          updateMin($colNum,minColNum,\psi_{0},\theta_{0},v_{0}$) in Algorithm 3;26:        **end if**27:     **end for**28:   **end for**29:**end for**30:Return $V_{c}^{I}=R\left(\psi \left(k\right),\theta \left(k\right)\right)*R\left(minTheta\right)*R\left(minPsi\right)*minV*u$ as the velocity command;

In Algorithm 4, $col(V_{c}^{I},objectSet)$ returns the number of objectPoints which collide with the velocity $V_{c}^{I}$. If colNum==0, then $V_{c}^{I}$ avoids colliding with every objectPoint in objectSet. Thus, we set $V_{c}^{I}$ as the velocity command at sampling-index k+1, and this algorithm is completed. On the other hand, if colNum≠0, then $V_{c}^{I}$ collides with at least one objectPoint in objectSet.

In Algorithm 4, maxColNum is set by considering the computational load of the algorithm. By setting maxColNum as a small number, one does not have to examine the non-collision requirement for every objectPoint in objectSet. In simulations, one uses maxColNum=100.
**Algorithm 3**$\;updateMin(colNum,minColNum,\psi_{0},\theta_{0},v_{0})$1:**if**colNum<minColNum; **then**2:   $minPsi=\psi_{0}$;3:   $minTheta=\theta_{0}$;4:   $minV=v_{0}$;5:   minColNum=colNum;6:**end if**


**Algorithm 4**

$\;col(V_{c}^{I},objectSet)$


1:colNum=0;2:**for** objectPoint $o_{p}\in objectSet$ **do**3:   **if** the velocity vector $V_{c}^{I}$ and $o_{p}$ do not satisfy the non-collision requirement; **then**4:     colNum=colNum+1;5:   **end if**6:   **if** colNum>maxColNum **then**7:     break;8:   **end if**9:
**end for**
10:Return colNum, and this algorithm is finished;


#### Find a Non-Collision Velocity Vector at Sampling-Index k+1 (Change Both Yaw and Pitch)

In the case where a non-collision velocity is not found under Algorithm 1, then Algorithm 2 runs to find a non-collision velocity command at sampling-index k+1, while AUV changes both its pitch and yaw. Since AUV’s pitch varies using Algorithm 1, the depth of AUV changes using this algorithm. In Algorithm 2, ($\theta_{0}=-\alpha *dt$, $\theta_{0}=\theta_{0}+\frac{2*\alpha *dt}{N}$, while $\theta_{0}<\alpha *dt$) is utilized to change the pitch of AUV.

Suppose that under Algorithm 2, one cannot find a non-collision velocity command, which avoids colliding with every objectPoint in objectSet. In this case, one finds a velocity command with the minimum collision probability by finding a velocity that collides with the minimum number of objectPoints. The velocity with the minimum collision probability is $V_{c}^{I}=R\left(\psi \left(k\right),\theta \left(k\right)\right)*R\left(minTheta\right)*R\left(minPsi\right)*minV*u$. See the last line of Algorithm 2.

### 5.3. AUV with a Designated Depth

Suppose that AUV has a designated depth. This designated depth can be useful, since we may not want the depth of the AUV to be too deep or too shallow. Moreover, suppose that there are many AUVs in the workspace. Then, we can set a distinct designated depth for every AUV. In this way, each AUV can move on distinct sea layer, while not colliding with other AUVs.

In Algorithm 2, ($\theta_{0}=-\alpha *dt$, $\theta_{0}=\theta_{0}+\frac{2*\alpha *dt}{N}$, while $\theta_{0}<\alpha *dt$) is utilized to change the pitch of AUV. Suppose that the designated depth is deeper than AUV’s current depth. In this case, ($\theta_{0}=-\alpha *dt$, $\theta_{0}=\theta_{0}+\frac{2*\alpha *dt}{N}$, while $\theta_{0}<\alpha *dt$) is utilized so that one examines a negative pitch before examining a positive pitch. Suppose that the designated depth is shallower than AUV’s current depth. In this case, ($\theta_{0}=\alpha *dt$, $\theta_{0}=\theta_{0}-\frac{2*\alpha *dt}{N}$, while $\theta_{0}>-\alpha *dt$) is utilized so that one examines a positive pitch before examining a negative pitch.

## 6. Controllers for Moving towards the Goal

We address how to set AUV’s velocity command so that it approaches the goal, while avoiding collision with obstacles. Let G denote the goal of AUV such that AUV needs to arrive at the goal safely. Using (Equation 4), let v(k) denote the $L^{2}$ norm of v(k), i.e., v(k)=∥v(k)∥. Moreover, let $G_{q}=G-q\left(k\right)$ for convenience.

AUV increases its speed so that it can reach the goal as fast as possible. This implies that one increases the AUV’s speed to $v_{d}$, which is given as follows.
(18)$$\begin{array}{c} v_{d}=min\left(v\left(k\right)+dt*a_{max},s_{max}\right). \end{array}$$

Here, the min operator is utilized, since $\left(v\left(k\right)+dt*a_{max}\right)$ cannot be larger than the AUV’s maximum speed $s_{max}$.

Suppose the following.
(19)$$\begin{array}{c} A(v\left(k\right),G_{q})<\alpha *dt. \end{array}$$

In this case, AUV’s velocity (v(k+1) in (Equation 4)) is set as follows.
(20)$$\begin{array}{c} v_{g}=\frac{v_{d}*G_{q}}{∥G_{q}∥}. \end{array}$$

Using $v_{g}$ in (Equation 20), the AUV heads towards the goal.

In the case where $v_{g}$ in (Equation 20) satisfies the non-collision requirement in Section 5.1, $v_{g}$ is utilized as the velocity of the AUV for a single sample duration. Using the definition of $G_{q}$, $v_{g}$ is set so that the AUV moves towards the goal.

See Figure 1 for an illustration. The sphere is centered at q(k), and its radius is ∥v(k)∥. The AUV is at q(k), which is the center of the sphere. v(k) is depicted as the bold arrow. $G_{q}$ is depicted as the dash dotted arrow. As the AUV at q(k) moves in the direction of $G_{q}$, it can head towards the goal of G.

In the case where $v_{g}$ does not satisfy the non-collision requirement, we use Algorithm 1 to find a velocity vector for avoiding collisions with obstacles. As long as $v_{g}$ satisfies the non-collision requirement, $v_{g}$ allows the AUV to reach the goal.

Suppose that
(21)$$\begin{array}{c} A(v\left(k\right),G_{q})\geq \alpha *dt \end{array}$$
holds. In this case, the AUV at q(k) cannot set $v_{g}$ in (Equation 20) as its velocity command due to the hardware limitations in (Equation 6). Therefore, the AUV turns with its maximum acceleration so that its velocity is as close to $v_{g}$ as possible.

Suppose that (Equation 21) holds. The AUV sets its velocity (v(k+1) in (Equation 4)) as follows:(22)$$\begin{array}{c} v_{d}=v_{d,t}+v_{d,n}. \end{array}$$

Here, $v_{d,t}$ is parallel to v(k), and $v_{d,n}$ is normal to v(k). $v_{d}$ is set so that AUV turns with its maximum acceleration such that its velocity is as close to $v_{g}$ as possible. We acknowledge that $v_{d}$ in (Equation 22) cannot assure that the AUV reaches the goal, since the AUV’s acceleration is bounded above.

In Figure 1, $v_{d}$ is depicted as the red arrow. Figure 1 illustrates the following:(23)$$\begin{array}{c} A(v\left(k\right),v_{d})=\alpha *dt. \end{array}$$

Figure 1 further illustrates that as the AUV at q(k) moves with velocity $v_{d}$, the AUV approaches the goal G.

In (Equation 22), one uses the following:(24)$$\begin{array}{c} v_{d,t}=v_{d}*c\left(\alpha *dt\right)\frac{v(k)}{v(k)}, \end{array}$$
which is parallel to v(k). Moreover, we have the following:(25)$$\begin{array}{c} v_{d,n}=v_{d}*s\left(\alpha *dt\right)\frac{G_{q}-G_{q}^{pro}}{∥G_{q}-G_{q}^{pro}∥}, \end{array}$$
where $G_{q}^{pro}$ is the projection of $G_{q}$ onto v(k) and is given by the following.
(26)$$\begin{array}{c} G_{q}^{pro}=\frac{(G_{q}\cdot v\left(k\right))v\left(k\right)}{{\left(v\left(k\right)\right)}^{2}}. \end{array}$$

In Figure 1, $G_{q}^{pro}$ is depicted as the dotted arrow. Moreover, $G_{q}-G_{q}^{pro}$ is depicted as the dashed arrow. As depicted in Figure 1, $\frac{G_{q}-G_{q}^{pro}}{∥G_{q}-G_{q}^{pro}∥}$ is normal to v(k). This further implies that $v_{d,n}$ in (Equation 25) is normal to v(k). The subscript *n* in $v_{d,n}$ in (Equation 22) implies that $v_{d,n}$ is normal to v(k), and the subscript *t* in $v_{d,t}$ in (Equation 22) implies that $v_{d,t}$ is tangential to v(k).

Using (Equation 22), (Equation 24), and (Equation 25), one satisfies the following.
(27)$$\begin{array}{c} v_{d}=∥v_{d,t}+v_{d,n}∥=∥v_{d}∥ \end{array}$$

Figure 1 illustrates that $v_{d}>v\left(k\right)$ using (Equation 18).

In the case where $v_{d}$ in (Equation 22) satisfies the non-collision requirement in Section 5.1, $v_{d}$ is utilized as the velocity of AUV for a single sample duration. However, in the case where $v_{d}$ does not satisfy the non-collision requirement, we use Algorithm 1 to find a velocity vector for avoiding collisions with obstacles.

## 7. MATLAB Simulations

The performance of the proposed controllers in Algorithm 1 is demonstrated by utilizing MATLAB computer simulations. We use MATLAB 2010 on the computer with the following specification: Intel(R)Core(TMi5−7600KCPU@3.80GHz).

The sample duration is dt=1 second, and $k_{p}=80$ is utilized in the reactive collision evasion control. The parameters associated with the ICP algorithm are $Th_{ICP}=10$ and $k_{d}=2$.

The AUV starts from (0,50,10) in meters. The process model of the AUV appeared in (Equation 4). In practice, AUV’s motion is perturbed by process noise, such as ocean currents. Considering this process noise, we use the following:(28)$$\begin{array}{c} q\left(k+1\right)=q\left(k\right)+v\left(k\right)*dt+\left(\begin{array}{c} rand\\ rand\\ rand \end{array}\right)*n_{p}-n_{p}/2, \end{array}$$
instead of (Equation 4). In (Equation 28), rand generates a random number between 0 and 1. $n_{p}$ indicates the process noise strength in (Equation 28). In MATLAB simulations, we use the $n_{p}=1$ meter by considering the drift due to ocean currents.

The goal of AUV is (0,500,50) in meters. AUV approaches the goal while evading collision with obstacles. To approach the goal, AUV uses the control law in Section 6.

AUV’s initial speed, maximum speed, maximum speed rate, and maximum turn rate are 6.3 m/s, $s_{max}=12.75$ m/s, $a_{max}$ = 2 m/s${}^{2}$, and α = 1.2 rad/s, respectively. Moreover, the minimum speed of the AUV is set as $s_{min}=1.2$ m/s. Note that the AUV needs to move along a narrow tunnel in simulations. The tunnel width is only 50 m.

We simulate the case where the AUV uses forward-looking sonar sensors for collision evasion. The maximum sensing range of the sonar sensors is 120 m. The sensor has 180 degrees in horizontal scan, such that 10 rays are evenly emanated in the horizontal direction. Moreover, it has 180 degrees in vertical scan, such that 10 rays are evenly emanated in the vertical direction. This implies that 10 × 10 sonar rays are emanated in total. As the number of sonar rays increases, we can have denser objectPoints on the obstacle boundary, which improves the performance of our collision avoidance controls. Our simulations use only 10 × 10 sonar rays, considering the hardware limit of AUV.

As a method to provide a realistic simulation environment, sonar sensor rays generate noisy position measurements. Recall that S(k) defines the set of objectPoints measured at sampling-index *k* and that S(k) is given by $S\left(k\right)=\left\{x_{i}\right\}$, $i\in \{1,\ldots ,M_{k}\}$. In MATLAB simulations, we use the following.
(29)$$\begin{array}{c} x_{i}=x_{i}^{p}+\left(\begin{array}{c} randn\\ randn\\ randn \end{array}\right)*n_{r}. \end{array}$$

Here, $x_{i}^{p}$ indicates an objectPoint derived using sonar rays without measurement noise. $x_{i}^{p}$ is derived by calculating the intersection between a sonar ray and an obstacle surface. In (Equation 29), randn generates a Gaussian noise with mean zero and variance 1. Moreover, $n_{r}$ indicates the standard deviation for the measurement noise in sonar rays. In MATLAB simulations, we use $n_{r}=0.05$ m.

As another method to provide a realistic simulation environment, sonar sensor rays are emanated with a low detection rate of 0.95. In other words, one randomly chooses 95 percent of all 10 × 10 sonar rays and utilizes the chosen rays to sense an obstacle boundary. As a chosen sonar ray intersects a boundary, an objectPoint is generated at the intersection point.

Recall we say that the AUV is in the near-collision state, in the case where the distance between AUV and any objectPoint is shorter than *r*. In simulations, r=10 is set by considering the size of AUV. In the case where the relative distance between AUV and an obstacle boundary is less than $r_{c}=5<r=10$ m, we assume that a collision has occurred. We use $r_{c}<r$ for AUV collisions, since sonar measurement error and the process noise can lead to the worst case where AUV enters the near-collision state. In other words, $r-r_{c}$ is the safety margin for the AUV.

In MATLAB simulations, the objectPoints measured by the AUV’s sonar sensors are illustrated with black points. In addition, the AUV’s path is illustrated with red circles. The goal is illustrated with the green asterisk. The simulations use one cylindrical obstacle and multiple box-shaped obstacles. The blue points in the figures indicate the vertices of each obstacle.

### 7.1. Scenario 1

We consider Scenario 1 with static obstacles. Figure 2 depicts the top view of the workspace. In addition, Figure 3 depicts the three-dimensional view of the workspace. These figures depict that AUV arrives at the goal (green asterisk) safely. See that objectPoints measured by sonar sensors are generated on obstacle surfaces.

Figure 4 depicts the state (speed, yaw, and pitch) of AUVs as time elapses, in the scenario of Figure 2. See that the AUV’s speed is upper bounded by $s_{max}$. The last subplot in Figure 4 shows the obstacle distance (distance from AUV to the closest obstacle) measured by AUV’s sonar. Since the maximum sensing range of the sonar sensors is 120 m, the obstacle’s distance is always less than 120 m. Initially, the AUV measures a cylindrical obstacle (circle located at (0,200) in Figure 2) and initiates changing its yaw for collision evasion. See that the AUV varies its state (speed, yaw, and pitch), while it moves.

Using MATLAB, it takes 44 s to run the entire simulation in Scenario 1. This computational time includes the time consumed to generate virtual obstacle environments, while we run the entire MATLAB simulation.

### 7.2. Scenario 2

We consider Scenario 2, in which the cylindrical obstacle in Figure 2 maneuvers with velocity (−3,−3,0) in m/s. At every 10 s, the moving obstacle (top and bottom circles of the cylindrical obstacle) is plotted. Figure 5 plots the top view of the workspace. Moreover, Figure 6 plots the three-dimensional view of the workspace. AUV arrives at the goal (green asterisk) while satisfying collision evasion.

Figure 7 depicts the state (speed, yaw, and pitch) of AUV as time elapses, considering the scenario in Figure 6. The last subplot in Figure 7 represents the obstacle distance (distance to the closest obstacle) measured by AUV’s sonar. Initially, the AUV measures a maneuvering cylindrical obstacle surface and changes its yaw for evading collision. See that the AUV varies its state (speed, yaw, and pitch) during the maneuver.

Using MATLAB, it takes 55 s to run the entire simulation in Scenario 2. This computational time includes the time spent to generate virtual obstacle environments, as one runs the entire MATLAB simulation.

### 7.3. Scenario 3

We consider Scenario 3, in which the cylindrical obstacle in Figure 2 maneuvers with velocity (0,−3,0) in m/s. At every 10 s, the moving obstacle (top and bottom circles of the cylindrical obstacle) is depicted. Figure 8 depicts the top view of the workspace, and Figure 9 plots the three-dimensional view of the workspace. These figures demonstrate that the AUV reaches the goal (green asterisk) while achieving collision evasion. The movie file of Figure 9 is uploaded on the following website: https://www.youtube.com/watch?v=WklOJ81QUsA (accessed on 22 July 2022 ).

Figure 10 depicts the state (speed, yaw, and pitch) of the AUV as time passes, considering the scenario in Figure 9. The last subplot in Figure 10 presents the obstacle distance (distance from AUV to the closest obstacle) measured by the AUV’s sonar. Initially, the AUV measures a maneuvering cylindrical obstacle surface and varies its yaw for evading collision. See that the AUV changes its state (speed, yaw, and pitch) during the maneuver.

Using MATLAB, it takes 41 s to run the entire simulation in Scenario 3. This computational time includes the time spent to construct virtual obstacle environments, while we run the entire MATLAB simulation.

## 8. Conclusions

This article introduced the reactive collision evasion controls, considering three-dimensional scenarios where AUV measures extended obstacles (moving obstacles as well as static obstacles) utilizing on-board sonar sensors. The velocity of each objectPoint of an extended obstacle is estimated using ICP algorithms. The estimated obstacle velocity is then utilized as the input for reactive collision evasion of the obstacle.

We further discussed how to make the AUV move towards the goal while avoiding collision with extended obstacles. The performance of the proposed reactive controllers was demonstrated utilizing computer simulations. In the future, we will demonstrate the performance of the proposed controls utilizing experiments with real underwater robots.

The proposed collision evasion controls can be integrated with trajectory tracking controls in the literature [32,33,34]. We derive the position and orientation of the AUV η=[x,y,z,ϕ,θ,ψ] at each sampling-index *k* [34]. Here, [x,y,z] presents the position coordinates, and [ϕ,θ,ψ] presents the orientation of AUV. In $\dot{\eta }$ at sampling-index *k*, $\left[\dot{x},\dot{y},\dot{z}\right]$ presents the velocity command v(k) in (Equation 4). At each sampling-index *k*, the AUV examines whether the calculated command v(k) meets the non-collision requirement. If the non-collision requirement is met, then the AUV uses the command v(k) for tracking the trajectory. Otherwise, the AUV uses the proposed collision evasion controls for avoiding collision with obstacles.

This article considers reactive control for a single AUV. In the future, we will consider path planning of multiple AUVs while evading collision simultaneously. References [36,37,38,39] can be utilized for localization of multiple AUVs. Moreover, Refs. [40,41,42,43] can be utilized for the reliable formation control of multiple AUVs.

## Figures and Tables

**Figure 1 sensors-22-05478-f001:**
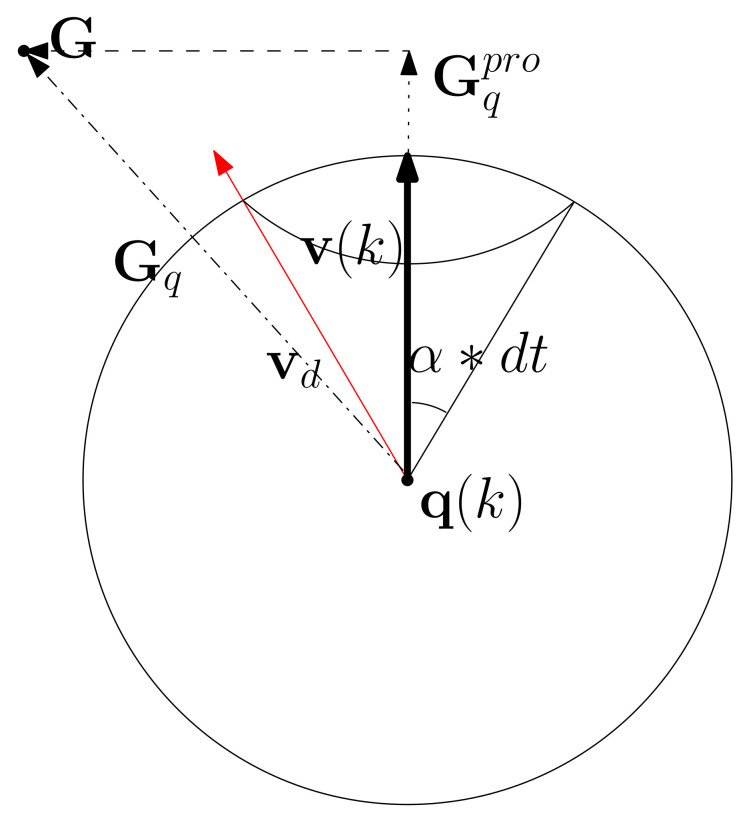
The sphere is centered at q(k), and its radius is ∥v(k)∥. The AUV is at q(k), the center of the sphere. $G_{q}$ is depicted as the dash dotted arrow. As the AUV at q(k) moves in the direction of $G_{q}$, it can head towards the goal.

**Figure 2 sensors-22-05478-f002:**
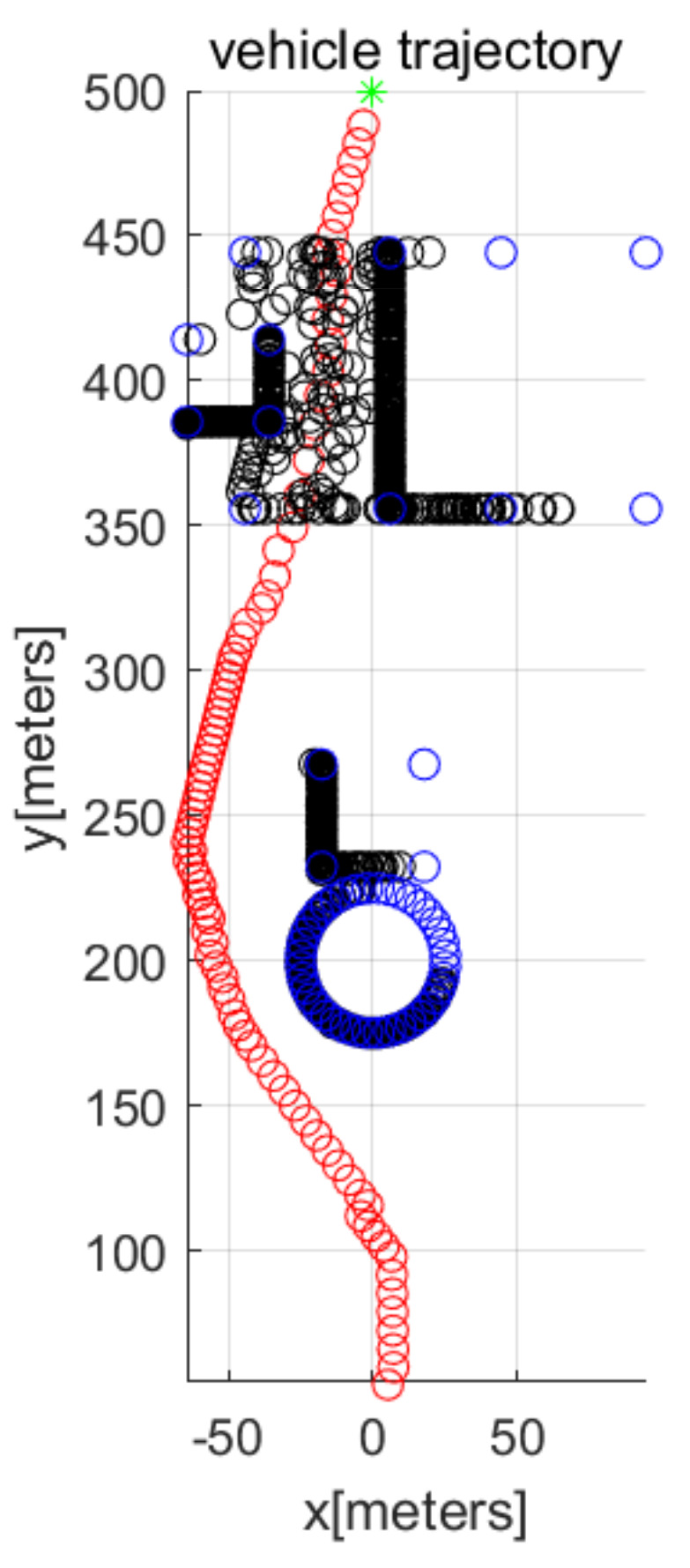
Scenario 1 with static obstacles. AUV starts from (0,50,10) (top view).

**Figure 3 sensors-22-05478-f003:**
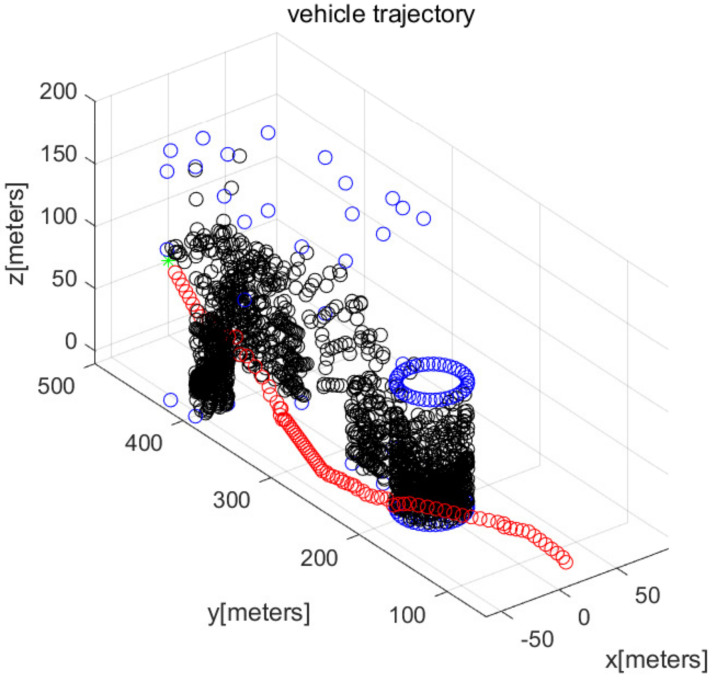
Scenario 1 with static obstacles. AUV starts from (0,50,10) (three-dimensional view).

**Figure 4 sensors-22-05478-f004:**
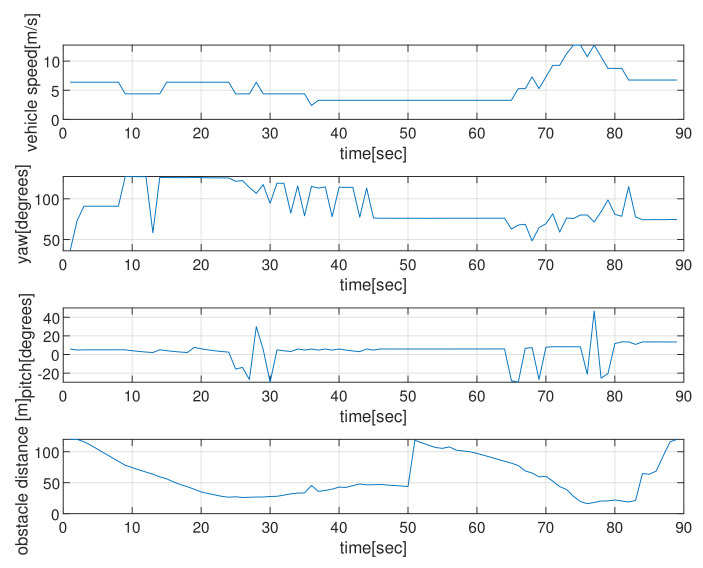
Scenario 1 with static obstacles. The state (speed, yaw, and pitch) of the AUV as time elapses, in the scenario of Figure 2. The last subplot in Figure 4 shows the obstacle’s distance (distance from the AUV to the closest obstacle) measured by AUV’s sonar. Since the maximum sensing range of the sonar sensors is 120 m, the obstacle’s distance is always less than 120 m.

**Figure 5 sensors-22-05478-f005:**
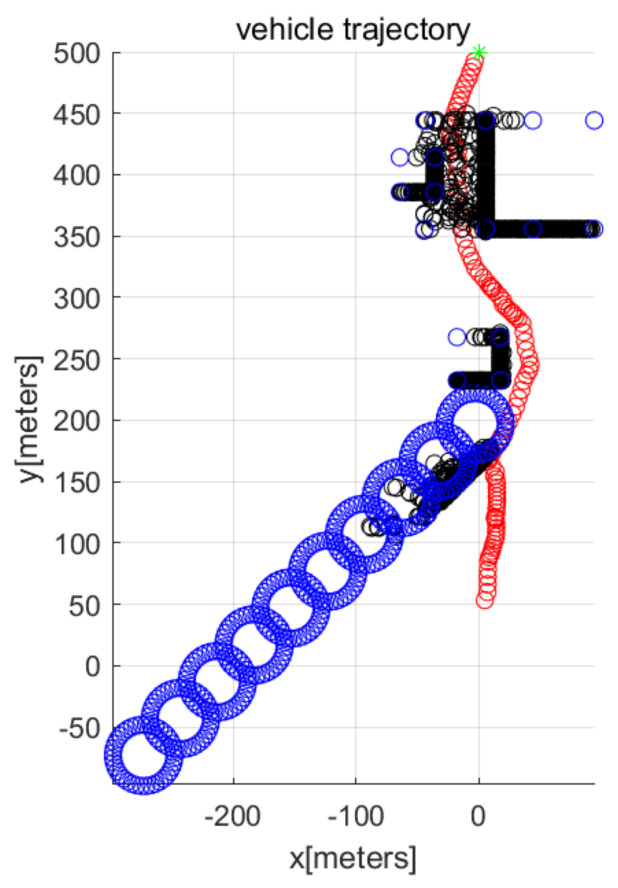
Scenario 2. The cylindrical obstacle maneuvers with velocity (−3,−3,0) in m/s (top view). At every 10 s, the maneuvering obstacle (top and bottom circles of the cylindrical obstacle) is depicted. The AUV reaches the goal (green asterisk) while assuring collision evasion.

**Figure 6 sensors-22-05478-f006:**
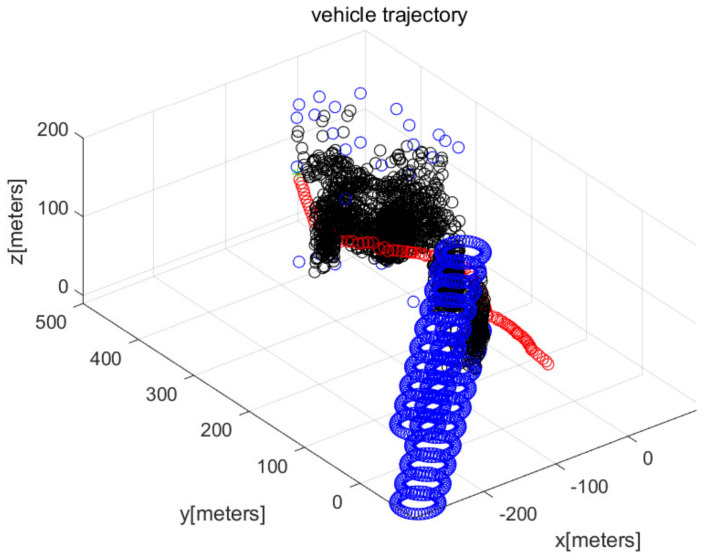
Scenario 2. The cylindrical obstacle maneuvers with velocity (−3,−3,0) in m/s (3D view). At every 10 s, the moving obstacle (top and bottom circles of the cylindrical obstacle) is plotted. The AUV arrives at the goal (green asterisk) while satisfying collision evasion.

**Figure 7 sensors-22-05478-f007:**
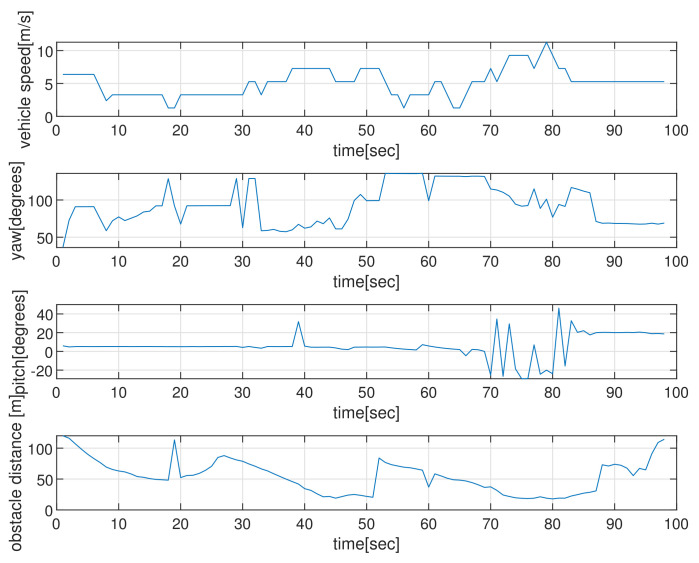
Scenario 2. The state (speed, yaw, and pitch) of the AUV as time elapses, considering the scenario in Figure 6. The last subplot shows the obstacle distance (distance to the closest obstacle) measured by AUV’s sonar.

**Figure 8 sensors-22-05478-f008:**
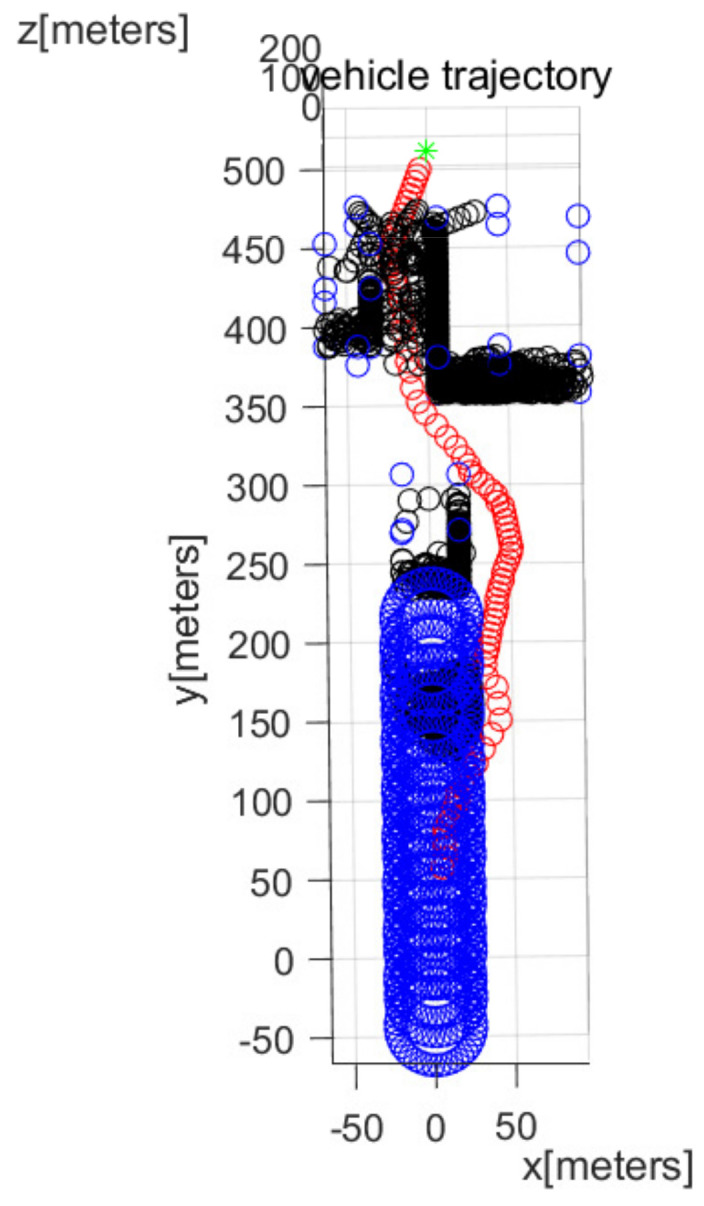
Scenario 3. The cylindrical obstacle maneuvers with velocity (0,−3,0) in m/s (top view). At every 10 s, the maneuvering obstacle (top and bottom circles of the cylindrical obstacle) is depicted.

**Figure 9 sensors-22-05478-f009:**
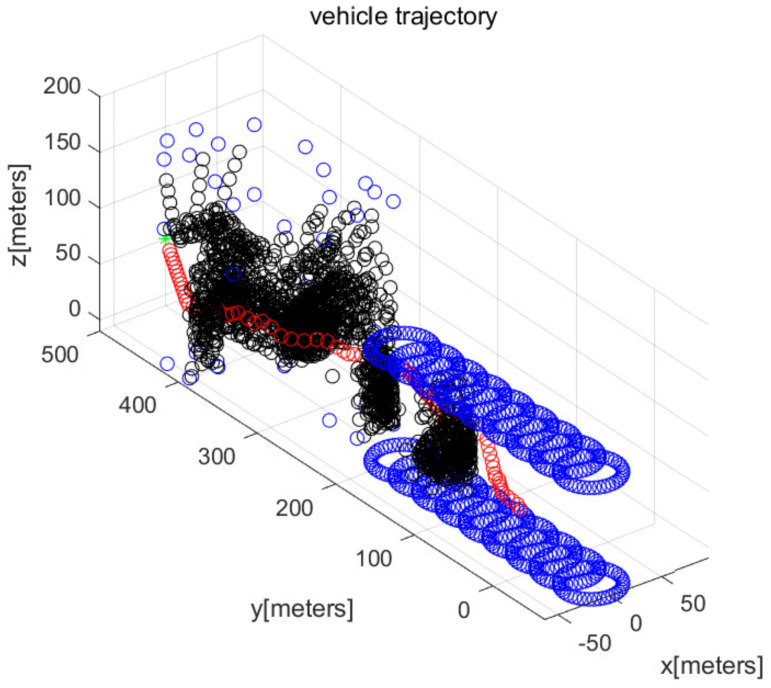
Scenario 3. The cylindrical obstacle maneuvers with velocity (0,−3,0) in m/s (three-dimensional view). At every 10 s, the moving obstacle (top and bottom circles of the cylindrical obstacle) is depicted.

**Figure 10 sensors-22-05478-f010:**
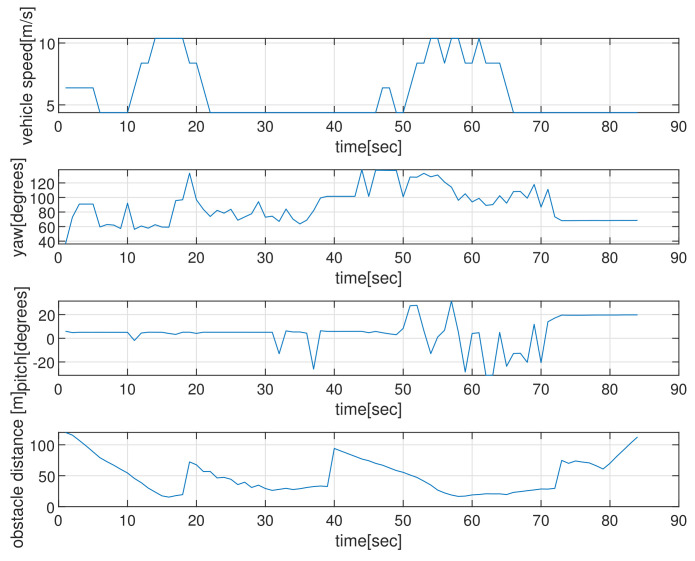
Scenario 3. The state (speed, yaw, and pitch) of the AUV as time elapses, considering the scenario in Figure 9. The last subplot shows the obstacle’s distance (distance to the closest obstacle) measured by AUV’s sonar.

## Data Availability

Not applicable.

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
