# Peer review of "Reactive Control for Collision Evasion with Extended Obstacles"

_sensors, 2022, doi:10.3390/s22155478_

Round 1
Reviewer 1 Report
This paper handles collision evasion with both moving obstacles and static obstacles with ICP algorithms. There are some major comments needed to be addressed before it could be considered for publication:
1. Please describe the methods in Section 3,4,5 and 6 in a clear and concise manner. For example, in the second paragraph of Section 3, there is no need to explain what min and max mean, because everyone knows. A scientific paper is not a story that requires you to recall all the time.
2. In Section 7, the authors use np=0.03m to simulate the actual error. Wheher it is too small for the actual marine environment? And the error is expressed with standard deviation, is it more reasonable to use the maximum and minimum to simulate the actual ocean current error?
3. Line 422 says that “Later, the AUV measures the box-shaped obstacle (located at (0,300)”, but I cannot see the obstacle as you said in fig.2.
4. As for the diagram of experimental results in Section 7, I am quite clear about what the symbols you used represent, but I still have difficulty in understanding some of the diagrams, such as fig 3, fig 5, fig 6 and fig 8. It is recommended to redraw the diagram to clearly represent obstacles and trajectories and describe in details in the text before discussing the results.
In addition, there are many typos and some grammar errors. There are some repetitive sentences in the introduction section, e.g., “Considering 3D environments, our paper handles collision evasion with moving obstacles as well as static obstacles.”, “To the best of our knowledge, the proposed collision evasion controllers are novel, since we handle the case where the AUV measures three-dimensional obstacle surfaces utilizing sonar sensors. ” Lastly, “The AUV” -> “AUV” throughout the paper.
Author Response
Thank you very much for your valuable comments. The response to Reviewer 1 is attached.

Reviewer 2 Report
Brief summary
This study proposes a AUV reactive collision evasion control. Its main contributions consist in considering collision avoidance with an extended obstacle, in the case where the AUV measures three-dimensional obstacle surfaces utilizing sonar sensors.
Broad comments
The document is relatively easy to read and follow.
The English needs minor review.
The document is well supported with references although the majority are old.
The subject of the paper is interesting and with a great potential of application.
One of the weaknesses of this study is the lack of experiments with real scenarios. Since the simulations do not include all the variables in order to reproduce real situations.
Specific comments
In line 57 please correct to “…thus appropriate for mobile robots”
In the introduction section there are many repetitions of “our paper”. Please rephrase.
The idea in line 70 is similar to line 55. Repetition of ideas. Please correct.
The text in line 86, 87 is the same as in line 102, 103. Please correct.
The Introduction section should be revised by the author taking also into account the above observations.
In line 130 did you mean “smallest value”?
In line 131 did you mean “biggest value”?
In line 165 author says that “…we reset our collision avoidance controls at each sampling-index”. Author should explain in the text what does exactly involve the reset of the collision avoidance controls.
Author states in line 171 that “AUV and an objectPoint op is in the near-collision state as the relative distance between them is less than a certain constant … set considering the size of the AUV.” This statement gives the idea that the distance is only calculated based on the AUV size. Shouldn’t this constant be calculated based also on the velocity of the vehicle and moving objects?! Please explain in the text.
Algorithms 1 thru 4 should be inserted closer to the text that refers each algorithm.
In line 426 please correct to “…time spent”.
Each figure should also appear closer to the first reference in the text.
In line 428, 443 and 457 author claim that “…AUV control is suitable for real time applications”. Author should explain in the text why, based on the obtained results, it concludes that AUV control is suitable for real time applications.
Author Response
Thank you very much for your valuable comments. The response to Reviewer 2 is attached.

Round 2
Reviewer 1 Report
The paper has been improved. The authors have resolved my concerns and revised the manuscript accordingly.